# Optimized Bioproduction of Itaconic and Fumaric Acids Based on Solid-State Fermentation of Lignocellulosic Biomass

**DOI:** 10.3390/molecules25051070

**Published:** 2020-02-27

**Authors:** Amparo Jiménez-Quero, Eric Pollet, Luc Avérous, Vincent Phalip

**Affiliations:** 1BioTeam/ICPEES-ECPM, UMR CNRS 7515, Université de Strasbourg, 25 rue Becquerel, 67087 Strasbourg, CEDEX 2, France; amparojq@kth.se (A.J.-Q.); eric.pollet@unistra.fr (E.P.); 2Division of Glycoscience, Department of Chemistry, School of Engineering Sciences in Chemistry, Biotechnology and Health, KTH Royal Institute of Technology, 10044 Stockholm, Sweden; 3Univ. Lille, INRA, ISA, Univ. Artois, Univ. Littoral Côte d’Opale, EA 7394—ICV—Institut Charles Viollette, F-59000 Lille, France; vincent.phalip@polytech-lille.fr

**Keywords:** lignocellulosic biomass, solid-state fermentation, enzymatic hydrolysis, aerated bioreactor, *Aspergillus oryzae*

## Abstract

The bioproduction of high-value chemicals such as itaconic and fumaric acids (IA and FA, respectively) from renewable resources via solid-state fermentation (SSF) represents an alternative to the current bioprocesses of submerged fermentation using refined sugars. Both acids are excellent platform chemicals with a wide range of applications in different market, such as plastics, coating, or cosmetics. The use of lignocellulosic biomass instead of food resources (starch or grains) in the frame of a sustainable development for IA and FA bioproduction is of prime importance. Filamentous fungi, especially belonging to the *Aspergillus* genus, have shown a great capacity to produce these organic dicarboxylic acids. This study attempts to develop and optimize the SSF conditions with lignocellulosic biomasses using *A. terreus* and *A. oryzae* to produce IA and FA. First, a kinetic study of SSF was performed with non-food resources (wheat bran and corn cobs) and a panel of pH and moisture conditions was studied during fermentation. Next, a new process using an enzymatic cocktail simultaneously with SSF was investigated in order to facilitate the use of the biomass as microbial substrate. Finally, a large-scale fermentation process was developed for SSF using corn cobs with *A. oryzae*; this specific condition showed the best yield in acid production. The yields achieved were 0.05 mg of IA and 0.16 mg of FA per gram of biomass after 48 h. These values currently represent the highest reported productions for SSF from raw lignocellulosic biomass.

## 1. Introduction

Solid-state fermentation (SSF) has emerged in the last decades as a promising industrial process for several products, especially using agricultural byproducts as the substrates [1,2]. SSF involves the growth of a microorganism on solid particles in the quasi absence of free water, and the majority of processes are performed by filamentous fungi under aerobic conditions [3]. The substrates used in SSF are often the source of nutrients for the microorganisms, and the inter-particle spaces allow gas and nutrients exchange between fungal hyphae and the medium. Fungi also behave as biocatalysts for the bioconversion of the substrates into specific target products such as bio-based fuels, commodity chemicals, enzymes, bioactive compounds, or food products [4].

SSF offers several advantages compared to submerged fermentation (SmF) such as high volumetric productivity, product concentration, simpler and smaller bioreactors because of the minimal free water, a lower sterilization cost, less generation of effluents (reduced cost of effluent treatment), and easier aeration due to lower density of the corresponding medium with high porosity [5,6]. Finally, the conditions of SSF mimic the natural environments of the filamentous fungi. However, the SSF process is slower compared to SmF, and all fermentation conditions cannot be controlled precisely. The main factors affecting fungal growth and metabolism in SSF are the selection of a suitable microorganism and substrate for the targeted generation of products, the pre-treatment of the substrate, the moisture, the temperature, and the removal of metabolic heat and gas transfers [7].

One of the most interesting biotechnological applications of SSF is the production of commodity chemicals [8,9]. The biosynthesis of chemicals from biomass creates a sustainable alternative to the conventional chemical synthesis based on fossil resources [10,11]. In the last two decades, many molecules produced from biomass with a large range of applications have been described [12,13]. Many of these building blocks are organic acids because of their capacities to generate high-value products for widespread industries such as food, pharmaceuticals or polymers [14,15,16]. The biosynthesis of by filamentous fungi has been studied extensively, and *Aspergilli* are often used for industrial production [17].

Among the organic acids, fumaric and itaconic acids (FA and IA, respectively) are included on the DOE’s (Unites State Department of Energy) list as part of the top twelve biomass-derived platform chemicals [12]. Both acids are polyfunctional building blocks that can be polymerized for instance, to give homo- or co-polymers for applications in textile, chemical, and pharmaceutical industries. IA can be used to replace acrylic acid, an important and rather costly chemical that is non-renewable so far, while FA can be used as food additive and in psoriasis treatment [14,16]. These acids are part of the tricarboxylic acid (TCA) cycle, FA being a direct intermediate in the cycle and IA a derivate of cis-aconitate acid, and both are produced under aerobic conditions. Currently, the industrial production of FA is via catalytic isomerization of fossil-based maleic acid. However, FA could also be produced biologically as an intermediate of the TCA cycle that is present in most aerobic organisms. Laboratory-scale fermentations with *Rhizopus oryzae* have shown interesting productivities in SmF of lignocellulosic biomass, around 0.35 g/g corn straw [18,19]. IA is produced industrially by *Aspergillus terreus* in SmF with glucose as the principal carbon and energy source to a yield of 100 g/L [20,21,22]. The biosynthesis involves the action of the cis-aconitate decarboxylase (CAD) enzyme to transform the cis-aconitate into itaconate. The presence of CAD in *A. terreus* has been demonstrated in different studies, but this enzyme is also present in another Aspergillus species; *A. oryzae* [23,24]. This aerobic, filamentous fungus is frequently used in SSF processes due to its capacity to hydrolyze the lignocellulosic substrates by enzymatic degradation [25]. Nevertheless, the production of IA and FA by SSF with lignocellulosic biomass has not been studied extensively in the literature. A method of SSF using sugarcane pressmud as a support for IA production, which yielded 0.0003 g kg^−1^ h^−1^, was patented in 2001 [26]. In this case, the main carbon source for the acid production has been added as liquid medium and the remaining sucrose for the sugarcane was a supplementary source. A maximum productivity of 0.021 g kg^−1^ h^−1^ of FA was reported via SSF of corn distiller grains by *R. oryzae* [27]. In a previous study, we have shown the capacity of *A. terreus* and *A. oryzae* to produce both acids by SSF process [28,29]. The yields obtained by *A. oryzae* from corn cobs were the most interesting, with 0.05 and 0.18 mg acid/g biomass of IA and FA, respectively. As expected, both productivities were lower than values reported from SmF processes utilizing soluble sugars in liquid media.

The aim of this work was to optimize the bioproduction of IA and FA in SSF by two *Aspergillus* species (*A. terreus* and *A. oryzae*) using lignocellulosic biomass as a non-food carbon source (wheat bran and corn cobs). As it is a complex and recalcitrant structure, lignocellulosic biomass is often predigested to be used as a sugar source for fermentation [30]. The bioconversion is carried out by enzymes produced under specific conditions by many microorganisms [31]. This saccharification process requires several hydrolytic enzymes such as cellulases, hemicellulases, xylanases, etc. [32]. The biomass pretreatment is one of the most expensive part of the lignocellulosic material conversion in an industrial scale. Therefore, the process could be coupled with the fermentation in a simultaneous saccharification–fermentation step, in order to improve the process yields. The study of this simultaneous process can open the possibility to decrease cost and time for an industrial activity in future. Several factors were studied to enhance organic acid production yields (pH, moisture content, enzyme hydrolysis) and large-scale fermentations were tested using these optimized factors.

## 2. Results

### 2.1. Solid-State Fermentation Kinetics

Fermentations were performed for both fungal species using both biomasses, wheat bran, and corn cobs. Organic acid production and fungal growth were studied. The determination of protein secretion level showed that *A. terreus* and *A. oryzae* present different development trends on wheat bran and corn cobs, Figure 1. The growth of *A. oryzae* on wheat bran reached a plateau after 120 h whereas *A. terreus* grew more slowly and regularly for more than 200 h on both substrates. Both species grew better on wheat bran than on corn cobs, as reported before in our previous studies, where both fungi species were capable of producing higher amounts of hydrolytic enzymes on wheat bran biomass [28,29]. These results were also confirmed by visual observations. More FA was produced from wheat bran with 0.8 and 0.6 mg/g biomass for *A. terreus* and *A. oryzae*, respectively, at the end of the fermentation with a regular increase in the yields. On corn cobs, FA production displayed a completely different profile, with a maximum yield after 48 h (0.14 and 0.12 mg/g for *A. terreus* and *A. oryzae*, respectively) and then a regular decrease.

Although the fungal growth was significantly higher on wheat bran, IA was produced only on corn cobs for both fungi, Figure 1. The different composition of both biomasses [28] may explain this behavior. A maximum IA yield of 0.025 mg/g corn cobs was produced by *A. oryzae* at 168 h of fermentation. At the same fermentation time, *A. terreus* produced half of this amount (0.012 mg IA/g biomass). The fungal biomass of *A. oryzae* was 15 times lower on corn cobs than on wheat bran and almost 2.5 times lower than the one of *A. terreus* on corn cobs.

### 2.2. Optimization of the SSF Steps

After testing the fermentations with both biomasses and fungi, the optimization steps were performed in two different ways. Firstly, a study with varying pH and humidity levels was carried out with corn cobs, to further improve the IA production. Secondly, an optimization of *A. oryzae* fermentation (displaying the highest IA production yield) was performed both on wheat bran and corn cobs by adding an enzyme cocktail to better hydrolyze the lignocellulosic biomasses. That could allow a more efficient conversion of accessible fermentable sugars in order to increase the yield of the fermentation products.

#### 2.2.1. Effect of pH and Moisture Level

Optimum pH and moisture level are crucial factors in SSF processes to obtain maximum yields of the products of interest [6,33]. The initial and previously tested conditions for corn cob fermentation (Section 2.1.) were pH 5 for the inoculation and 90% humidity. To optimize the pH and moisture conditions, five different pH values and five different moisture levels were evaluated for the inoculation step of the biomass culture, Figure 2 and Appendix A. 

For *A. terreus,* both acids were produced with higher yields at pH 6, as reported before [28,34]. The moisture content influenced the IA and FA production differently. The best IA production, 0.025 mg IA/g corn cobs, was observed at pH 6 and 130% humidity (Figure 2A) i.e., a doubling of the production compared to the initial conditions (pH 5 and 90% humidity). FA was produced at a yield of 0.095 mg/g biomass (pH 6 and 70% humidity), which is also almost twice the production at initial conditions (Figure 2B). For both acids, a clear trend is that a neutral pH (pH = 7) seems too high (Figure 2A,B). This observation is in good agreement with previous results obtained for SmF [31].

*A. oryzae* also showed a preference for pH 6, for the production of both acids (Figure 2C,D). In the case of IA, the highest yield was 0.045 mg/g biomass at 110% humidity (Figure 2C), which is slightly higher than the yield under the initial conditions (0.039 mg IA/g biomass) and almost twice the yield obtained with *A. terreus.* Under the same conditions (pH 6, 110% humidity), 0.091 mg FA/g was produced (Figure 2D). It is not the highest yield since at 130% humidity, the production was even higher (0.111 mg/g biomass).

#### 2.2.2. Enzymatic Hydrolysis

The enzymatic cocktail was obtained by SSF of wheat bran by *A. oryzae*, as shown in previous studies [29], the solid fermentation of the biomass allows to produce higher protein content with specific lignocellulolytic enzymes for biomass degradation. The cocktail can be store at −20 °C and used for simultaneous SSF.

The enzyme cocktail produced by *A. oryzae* showed the best enzymatic activity for endoxylanases (Table 1), which are responsible for hemicellulose hydrolysis. Hemicellulose is the most abundant part of corn cobs [35,36]. Moreover, cellulase and xyloglucanase activities were also found, suggesting an efficient biomass digestion. When the enzymatic cocktail was used for simultaneous saccharification-fermentation of corn cobs, *A. oryzae* rapidly secreted proteins (i.e., it grew) in the first 20 h, and a plateau was subsequently reached (Figure 3B) approximately at the same level as with the raw biomass (Figure 1D). Surprisingly, with the treated wheat bran (Figure 3A), the fungi secreted half of the proteins compared to the untreated biomass (Figure 1C). Moreover, the treatment had a dramatic negative effect on the production of FA from wheat bran (Figure 3A) with a yield (0.08 mg/g biomass) almost 8 times lower than without pretreatment (0.6 mg/g of biomass) (Figure 1C). In contrast, for corn cobs, the FA yield was feebly increased to 0.15 mg/g biomass. The profile of FA production from corn cobs (Figure 3B) was similar to the one without the enzyme cocktail (Figure 1D) with a yield reaching a maximum (after ca. 50 h) followed by a decay.

The best contribution of the enzyme cocktail was observed for the production of IA. According to our knowledge, the use of such enzymatic cocktail allowing IA production from wheat bran was reported for the first time in this study. IA production was detectable after 22 h, and a maximum yield of 0.046 mg/g biomass was obtained after 66 h (Figure 3A). With corn cobs, IA production is clearly detected earlier (14 h), and a yield of 0.052 mg/g biomass was achieved (Figure 3B) that was twice the maximum yield produced without enzymatic treatment for optimized pH and moisture level, Figure 2C.

### 2.3. Kinetics of SSF with Optimized Conditions

According to previous results, the optimum fermentative process was 80 h at pH 6 and 110% humidity. Figure 4 presents the glass flasks and shows the development of *A. oryzae* through the fermentation (with a green color indicating a high spore concentration). The fungus grew progressively during the fermentation until 0.22 mg of protein/g of biomass was obtained, as shown earlier (at pH 5 and 90% moisture level) at the same time of fermentation (Section 2.2.1). In relatively good agreement, FA production was only slightly enhanced (+ 10%) with a maximum yield of 0.16 mg/g biomass within 48 h (Figure 4). However, for IA, the enhancement was higher because the production was more than doubled (0.051 mg/g biomass) after 48 h and higher (0.061 mg/g) after 80 h (factor 2.4). As generally described for fungi, *A. oryzae* metabolism is greatly influenced by pH and the humidity level at the start of the fermentation step [37,38].

### 2.4. Larger Scale Fermentation

To develop and analyze the scaling up, the fermentation was performed with 200 g of corn cobs i.e., 20 times more than for the previous glass flasks experiments. The optimized conditions of pH and moisture (i.e., 110% moisture and pH = 6) were applied for the scaled-up fermentation. Aeration plays an important role in SSF for the transfer of oxygen and the evacuation of the carbon dioxide produced. Aeration is also used (Figure 5) to dissipate the metabolic heat generated by fermentation [39]. The mixture of substrates (solid lignocellulose particles and fungal mycelium) also helps to equilibrate the gas exchange, temperature, and moisture level [3], avoiding the disruption of the mycelial-substrate contact, which is particularly important for *A. oryzae*, for instance, to produce the degrading enzymes to hydrolyze the biomass.

Two different processes were used to test the influence of both aeration and mixing. A monolayer reactor presenting the same conditions as the glass flasks except for size was compared with an aerated plastic bag fermenter. The plastic bag fermenter was gently mixed on a rocker shaker and distilled water was added in a timely manner to equilibrate the moisture level. The fungal growth was clearly different between the two fermenters (Figure 6). The aerated plastic bag yielded nearly twice the proteins concentration (0.26 mg/g biomass) as the monolayer fermenter (0.15 mg/g biomass) at the end of the fermentation. Compared with the glass flask fermentation (0.22 mg protein/g biomass, Figure 4) the monolayer fermenter produced less protein whereas the aerated one displayed an amount of protein similar to the small-scale fermentation. This difference in fungal growth was also obvious in observing *A. oryzae* sporulation, which occurred earlier for the monolayer fermenter (until the second fermentation day) than for the aerated fermenter. This premature sporulation indicates that mycelial development was interrupted by inadequate conditions. The aeration and the loss of humidity correction increased the fungal development and also delayed the sporulation.

The FA production was not affected by the different conditions (0.09 mg/g biomass in both reactors), probably because 96 h of fermentation was not an optimized time to recover FA as shown in the glass flask (Figure 4) where the maximum FA yield was produced at 36 h. Conversely, IA production was 60% higher in the aerated fermenter (Figure 6) than in the monolayer reactor.

## 3. Discussion

From the SSF kinetics experiments, we can conclude that IA production was inversely linked to growth. Both acid yields were lower than the yields from the current SmF [31,32,33], then the optimization of the fermentative conditions is necessary to enhance the acid production. During the optimization steps, we could observe how low moisture content causes slower enzyme secretion from the fungus due to the lower solubility of the nutrients and the low level of growth [37,38] (Appendix A). However, acidic pH (3–5) and low moisture often allowed better production of the acids (Figure 2B,D). This behavior is not observed at higher pHs. These observations are in agreement with the fact that in SSF, pH variation significantly impacts the production and the stability of the enzymes [5,39], with several enzymes responsible for biomass hydrolysis during growth. The pH effect on the organic acid production was in accordance with the simultaneous fermentation and enzymatic hydrolysis study performed previously [29]. Even if the enzymatic cocktail did not improve the growth of the fungus nor FA production, the cocktail created better conditions to produce IA. FA is an intermediate metabolite of fungal fermentation, and its production is directly linked with the development of the microorganism unlike IA, a secondary metabolite [34]. The time lag between growth and IA production is perfectly consistent with a secondary metabolite behavior. These results showed the importance of a separated optimization of FA and IA production, as well as the time of recovery of the carboxylic acids during the fermentation, especially from a continuous production perspective.

The acid yields for the simultaneous saccharification and fermentation process were higher in comparison with the best results obtained for optimized pH and moisture (pH = 6 and 110% humidity) with corn cobs and *A. oryzae, (*Figure 2C,D). However, the production of the enzyme cocktail required four additional days for the entire process. Therefore, the most interesting strategy for the acid production was the kinetic fermentation of corn cob biomass by *A. oryzae* with the optimized conditions of pH and humidity. The kinetic curves showed the differential production over time for both acids, leaving the possibility to improve and optimize the time production of the acids if focusing exclusively on one of them. In this case, the pH and the moisture should be adapted from the previous results (Figure 2 and Appendix A).

The literature is deficient concerning SSF of FA and IA without biomass pretreatment. For IA, a mutant of *A. terreus* displays a productivity of 0.0003 mg/g h with sugarcane pressmud supplemented with sugars and nutrients [26]. The result obtained in this study was more than two times higher (0.00076 mg/g h) by *A. oryzae* (novel IA producer) with a lignocellulosic substrate without any nutrient addition [29]. Our study can contribute to further optimization on the fermentation conditions (pH, humidity, and aeration level), that combined with new studies using metabolic engineered microorganisms [40,41] providing better yields in IA production from lignocellulosic materials.

Regarding the larger scale fermentation in the plastic bag prototype reactor, the improvement in acid production could be explained by the better supply of oxygen and moisture in the fermenter. Indeed, IA fermentation is strictly aerobic, and previous studies showed that a gain in dissolved oxygen and agitation induced higher yields [42,43]. The moisture level could affect the IA production. Below 70% humidity, the nutrient transfers are limited, and the metabolism is affected [44]. In our experiment, water addition along with aeration may allow better fermentation conditions of the air-solids-water to enhance IA production. Most of these factors were studied for *A. terreus*, long known as an IA producer. However, for *A. oryzae*, conditions still need further optimization. Even if the IA yield obtained in the aerated fermenter (0.05 mg/g biomass) was similar to the small glass flask, the final production was multiplied by 20. Of course, the yields obtained in this work are still far from the industrial scale target, and further work in optimizing the fermentation conditions and down-stream processing have to be carried out. Anyway, *A. oryzae*, which showed the most interesting enzymes production for biomass degradation, seems to be an excellent candidate for further studies.

## 4. Materials and Methods

### 4.1. Feedstock and Microorganisms

Two agricultural waste biomasses used as non-food carbon sources were wheat bran and corn cobs obtained from Comptoir Agricole (Lauterbourg, France). The lignocellulosic material was milled (SX 100, Retsch) to obtain particles that were 0.5–1 mm in size. The water activity (Aw) was measured on 1 g of dry substrate by an Aw meter Fast-lab (GBX, France).

*A. terreus* (DSM 826) was provided by the Deutsche Sammlung von Mikroorganismen und Zellkulturen (DSMZ, Braunschweig, Germany). *A. oryzae* (UMIP 1042.72) was provided by the Fungal Culture Collection of the Pasteur Institute (France). The strains were revived on potato dextrose broth medium (PDB) for 5–6 days at 25 °C. The microorganisms were then grown and sporulated on potato dextrose agar (PDA). The spore suspensions were harvested from 5–6-day-old PDA plates with 0.2% (*v*/*v*) Tween-80. The spores were counted using a Malassez counting chamber and stored at −20 °C.

### 4.2. Initial SSF Step

In glass flasks, 5 g of solid substrate were autoclaved at 121 °C and 3 bars, for 20 min. The substrates were inoculated with spore suspensions to have an initial concentration of 10^6^ spores/g of substrate. Initial moisture was adjusted to 120% for wheat bran and 90% for corn cobs (both corresponding to an Aw near 1). After thorough mixing, the flasks were covered with a porous adhesive film (VWR, Radnor, PA, USA) and incubated at 30 °C for 6 days. Unless specified otherwise, these fermentation conditions were maintained throughout the study. All the experiments were conducted in duplicate. After fermentation, the samples were recovered by mixing the fermented substrate with sterilized distilled water (7 mL/g of initial dry substrate). The preparations were centrifuged (8000× *g* for 15 min) to eliminate residual solids. Then, another centrifugation step was performed on the supernatants to remove mycelia and spores (13,600× *g* for 30 min). The resulting solution was finally filtered through a 0.2 µm membrane. The samples were analyzed by high-performance liquid chromatography (HPLC, Waters, Milford, MA, USA) and were stored at −20 °C for additional analysis.

#### 4.2.1. pH and Humidity Level Optimization

The initial pH and humidity levels were varied to screen the best conditions for organic acid production. Citrate-phosphate buffer solutions, pH 3 to 7, were prepared. The moisture content was set at 50, 70, 90, 110, and 130 *v*/*w*. Five pH conditions were crossed with five relative humidity level, generating 25 different conditions performed with biological duplicates in SSF.

#### 4.2.2. Enzyme Production for Biomass Hydrolysis

The enzymatic cocktail preparation was performed in a plate with 250 g of wheat bran by *A. oryzae*. The inocula were prepared with 300 mL of Tris buffer at pH 10 to achieve a final spore concentration of 10^6^ per g of biomass, and the incubation temperature was 25 °C during the 4 days. The enzymatic cocktail was recovered with 1500 mL of sodium phosphate buffer at pH 6 and filtered with a Vivaflow 200 system (Sartorius, Göttingen, Germany). The cocktail was stored at 4 °C, and enzyme activities were determined.

### 4.3. Scale-Up Steps

Fermentation at a higher scale, with 200 g of biomass, was performed in two different types of reactor. A monolayer reactor consists of a glass plate covered with a gas exchange Miracloth film (Millipore, Temecula, CA, USA). The second reactor is a prototype of an aerated reactor (Figure 5). This fermenter was made from an autoclaved polypropylene laboratory bag with a central opening covered with a gas exchange film of Miracloth. Two PVC tubes were introduced and connected to stone ceramic air diffusers (3 mm in diameter). Another PVC tube was added for inoculation and double distilled water addition. The entire reactor was autoclaved with the biomass inside. During fermentation, double distilled water was added at the rate of 11 mL per day to keep the moisture constant (considering 10% evaporation/day). The aeration was provided by an air pump AC-9906 (Resun, Shenzhen, China) at a flow rate of 840 L/hour. A rocker mixer (ThermoFisher, Waltham, MA, USA) was used to shake the fermenter in order to uniformly add the water to the biomass. The influence of these controlled conditions of air and moisture content on the acid production could be studied for the plastic bag reactor but not for the monolayer reactor.

### 4.4. Analytical Procedure

#### 4.4.1. Mycelial Growth (Protein Assays)

To determine the fungal proteins produced during the fermentation, Bradford method was used [45]. All the samples were centrifuged and filtered (0.22 µm) before analysis to eliminate the spores. The protein assay was calibrated using BSA (bovine serum albumin) as the standard.

#### 4.4.2. Organic Acid Assays

A chromatographic system based on a 616 pump, a 2996 photodiode array detector operating in a range of 200 to 450 nm, and a 717 Plus autosampler (Waters, Milford, MA, USA) controlled by Empower 2 software (Waters, Milford, MA, USA) was used to analyze the samples, as previously described [29]. The columns were calibrated using commercial IA and FA samples with a 99.9% purity (Sigma-Aldrich, San Luis, MO, USA) and a UV measurement at 205 nm. Each sample was supplemented with 10 ppm IA or FA as the internal standard to confirm the acid production.

#### 4.4.3. Enzyme Activity Assay

Chromogenic substrates, azurine-crosslinked (AZCL) polysaccharides such as AZCL-HE-cellulose, AZCL-xylan, AZCL-xyloglucan, or AZCL-amylose (Megazyme, Bray, Ireland), were used to measure the enzyme cocktail activity. The samples were collected and analyzed as previously described [28], by spectrophotometry determining absorbance of the supernatant at 595 nm that corresponds with the solubilization of dyed compounds (AZCL) by the enzymes present in the cocktail.

## 5. Conclusions

The IA production process appears to be ideally amenable to SSF conditions, as demonstrated in this work. However, the fermentation conditions still need further optimization to provide yields similar to the yields obtained by submerged fermentation, considering the use of lignocellulosic substrates. Additionally, the use of a novel species, *A. oryzae* (which is used industrially for enzyme production) opens up the possibility of creating a biorefinery process for the production of both organic acids and enzymes. The use of agricultural wastes and cheap and non-food substrates in the bioprocess could lower IA production costs and could therefore promote the use of bio-based IA in the polymerization process to replace petroleum-derived polymers. Furthermore, the simultaneous production of another organic acid as FA by *Aspergillus* species can open the possibility to adapt the use of different lignocellulosic biomass for particular building blocks production. Moreover, in the case of downstream purification processes for both IA and FA, the opportunity of co-polymerization could be an interesting case of study. Anyway, the results showed that time production as well as the fungi needs can differ between IA or FA production, and specific individual optimization for SSF should also be done. Moreover, further studies need to be performed in the use of commercial enzymatic cocktails to improve the production yields, but also to understand the need for better biomass hydrolysis in SSF. In this sense, *A. oryzae* seem to be a great candidate for commercial enzymes production. One possibility could be the use of metabolic engineering to guaranty the strong lignocellulolytic enzyme release combined with specific metabolic route for IA or FA, exclusively. In this work, the organic acids yields were slightly lower than those obtained in previous studies. However, the global time-lapse of the process was greatly decreased considering that no previous pretreatment steps were preformed, or enzymatic cocktails were collected.

## Figures and Tables

**Figure 1 molecules-25-01070-f001:**
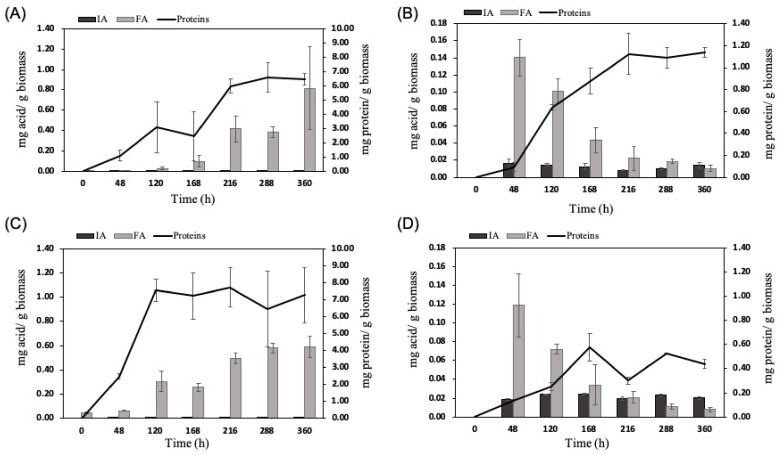
Fermentation kinetics on lignocellulosic biomasses: itaconic acid (IA) and fumaric acid (FA) yields and protein production (fungal growth) from wheat bran (**A**,**C**) and corn cobs (**B**,**D**) by *A. terreus* (**A** and **B**, respectively) and by *A. oryzae* (**C** and **D**, respectively).

**Figure 2 molecules-25-01070-f002:**
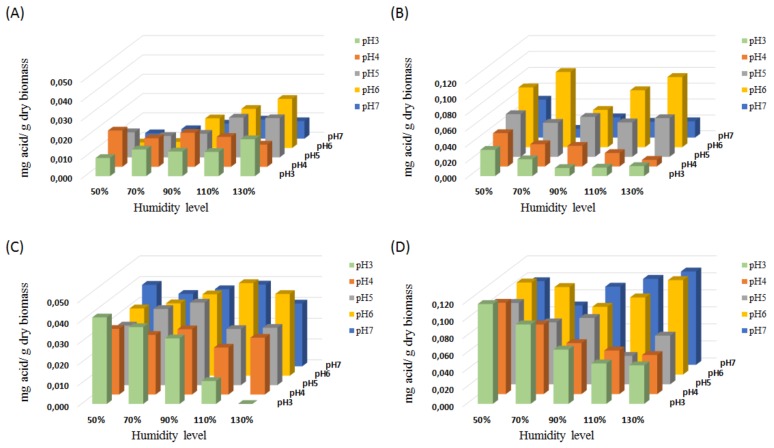
Solid-state fermentation (SSF) on corn cobs at different pH and moisture levels by *A. terreus* (IA and FA yields: **A** and **B**, respectively) and *A. oryzae* (**C** and **D**, respectively).

**Figure 3 molecules-25-01070-f003:**
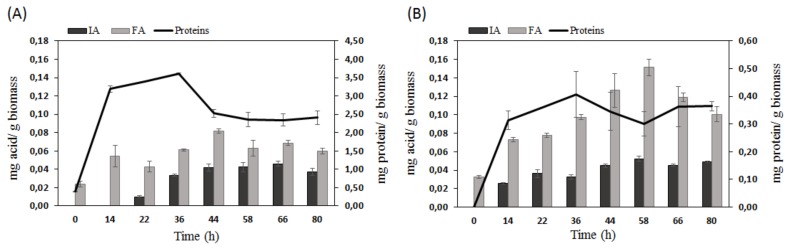
Kinetics of simultaneous saccharification and fermentation of wheat bran (**A**) and corn cobs (**B**) by *A. oryzae*.

**Figure 4 molecules-25-01070-f004:**
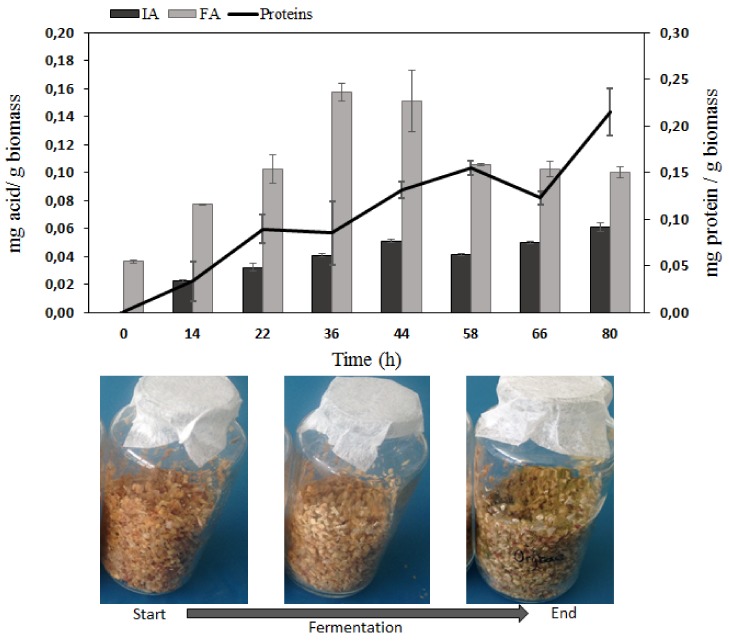
Kinetics of SSF on corn cobs by *A. oryzae* under optimized conditions (pH 6 and 110% moisture).

**Figure 5 molecules-25-01070-f005:**
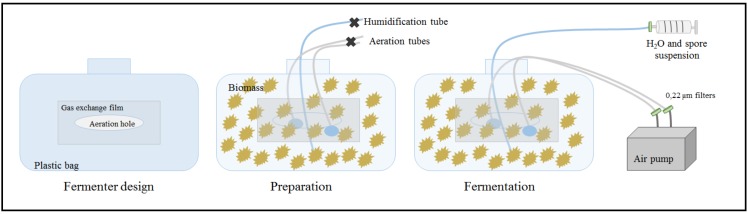
Illustration of aerated plastic bag fermenter (at left), made from autoclavable biohazard bags in polypropylene with an aeration hole covered by a gas exchange Miracloth film (Millipore, USA). Prior to autoclaving (middle), corn cobs were introduced as well as the air and humidification tubes (autoclavable tubes in PVC used for the liquid bioreactor). Right: Operative fermenter with aeration and the inoculum to be injected.

**Figure 6 molecules-25-01070-f006:**
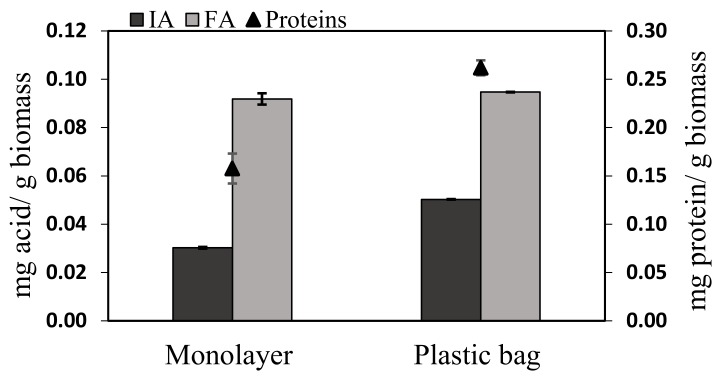
SSF in larger scale fermenters of *A. oryzae* from corn cobs: Organic acid productions and protein secretion (growth).

**Table 1 molecules-25-01070-t001:** Enzymatic activities (in ΔOD/g*min*) of the enzymatic cocktail obtained from SSF of wheat bran by *A. oryzae*.

α-Amylase activity	18.10
Cellulase activity (cellulose)	4.69
Endoxylanase activity	70.30
Cellulase activity (xyloglucan)	10.11

Enzyme activities were expressed in arbitrary units corresponding to optical density variations (ΔOD) per minute and per gram of biomass, due to the unknown extinction coefficient of AZCL substrate (Megazymes, Ireland).

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
