# Peer review of "Optimized Bioproduction of Itaconic and Fumaric Acids Based on Solid-State Fermentation of Lignocellulosic Biomass"

_molecules, 2020, doi:10.3390/molecules25051070_

Round 1

Reviewer 1 Report

In this manuscript, optimization of simultaneous production of itaconic and fumaric acids was conducted by solid-state fermentation of A. terreus and A. oryzae. Two lignocellulosic biomass (wheat bran and corn cobs) were used as carbon sources. Operation factors, such as initial pH, initial moisture content and biomass hydrolysis by enzyme cocktails, were tested with 5 g of solid substrate in glass flasks. Then, fermentation of 200-g-scale was also carried out.

These references were misquoted. In Ref. 26, sugarcane pressmud was used as support, not as substrate, to adsorb liquid medium for solid-state fermentation. In Ref. 27, fumaric Acid was produced by liquid-state fermentation.

Is the unit of yield based on dry weight of biomass? Productivity, mg acid/g dry biomass/ hr, may be an another good index in solid-state fermentation. Because it is not easy to analyze how much substrate was used.

The scales of Y-axis in Fig 1 were inconsistent. Yields of itaconic acids Production under different conditions are not easy to compare especially.

Simultaneous production of itaconic and fumaric acids might be not necessarily a good strategy, because production conditions were different and downstream processing became complicated. High-yield production conditions of itaconic and fumaric acids could be established, respectively.

Itaconic and fumaric acids did not produce at the same time. In addition to the factors of pH and moisture level, fermentation time should be added in experiments. Kinetics of larger scale fermentation is also required.

The method of plastic bag for solid-state fermentation would be described more detail. Why did the plastic bag need a shaker for substrate mixing? Was distilled water added uniformly in plastic bag? How was the gas exchange film installed in the plastic bag? How much is the air flow rate?

Reviewer 2 Report

The authors optimized itaconic and fumaric acid production from corn cobs and wheat bran, but the data presented completely lacks statistical analysis.

The topic is not new, nevertheless, it is of some interest as it’s the first report where A. oryzae is used for Itaconic acid production. Lack of some information in the methodology and the typos are below.

The descriptive form of the discussion does not allow to determine what results according to the authors are a priority, and which are of secondary importance. Pointing that out will be helpful to draw conclusions.  again discussion should be improved.

What are the differences between this manuscript and their published article (J. Microbiol. Biotechnol. (2017), 27(1), 1–8; 8
https://doi.org/10.4014/jmb.1607.07057)?  

Minor comments:

Line 35: change to "substrates"

Line 59: discuss briefly applications of IA and FA.

Line 66: how much yields (be quantitative)

Line 10: What could be the possible reason for this?

Figure 2: Where are error bars?

Line 134: Was it significantly higher at pH 6?

Table 1: Present in IU (Units per ml)

Line 220: Autoclave: it could act as a pretreatment

Line 255: "This point" - what point is this?

Line 267: How about compare your results with pretreatment studies 

Round 2

Reviewer 1 Report

In this study, the yields of itaconic acid and fumaric acid (line 28) were not higher than that of previous study (line 85). Moreover, we can find that the yield of solid fermentation was much lower than that of liquid fermentation, so the benefits of itaconic acid and fumaric acid production by solid fermentation need to be further evaluated.

Reviewer 2 Report

I would like to see statistical analysis for Figure 2 data.
